# Evaluating Mathematical Concordance Between Taxonomic and Functional Diversity Metrics in Benthic Macroinvertebrate Communities

**DOI:** 10.3390/biology14060692

**Published:** 2025-06-13

**Authors:** Gonzalo Sotomayor, Henrietta Hampel, Raúl F. Vázquez, Christine Van der heyden, Marie Anne Eurie Forio, Peter L. M. Goethals

**Affiliations:** 1Facultad de Ingeniería, Universidad de los Hemisferios, Paseo de la Universidad No. 300, Quito 170147, Ecuador; 2Laboratorio de Ecología Acuática (LEA), Facultad de Ciencias Químicas, Universidad de Cuenca, Av. Victor Albornoz y Calle de los Cerezos, Cuenca 010207, Ecuador; hennihampel@gmail.com (H.H.); raulfvazquezz@yahoo.co.uk (R.F.V.); 3Department of Animal Sciences and Aquatic Ecology, Faculty of Bioscience Engineering, Ghent University, Coupure Links 653, 9000 Ghent, Belgium; marie.forio@ugent.be (M.A.E.F.); peter.goethals@ugent.be (P.L.M.G.); 4Departamento de Ingeniería Civil, Facultad de Ingeniería, Universidad de Cuenca, Av. 12 de abril S/N, Cuenca 010201, Ecuador; 5Health and Water Research Center, Faculty of Science and Technology, Hogeschool Gent/University of Applied Sciences and Arts HOGENT, Valentin Vaerwyckweg 1, 9000 Ghent, Belgium; christine.vanderheyden@hogent.be

**Keywords:** mathematical concordance, functional diversity metrics, taxonomic diversity metrics, benthic macroinvertebrates, fluvial habitat quality gradient

## Abstract

Freshwater ecosystems are increasingly threatened by habitat degradation and other environmental pressures. To assess the condition of rivers and streams, scientists often use bioindicators such as aquatic macroinvertebrates. These organisms can be studied using traditional taxonomic approaches (based on taxa composition) or through functional traits (which describe their ecological roles). In this study, we analyzed data from the Paute River Basin situated in the tropical Andes of Ecuador to evaluate how well taxonomic and functional diversity metrics align with each other. Our results show that certain functional and taxonomic metrics provide similar ecological information, while others diverge. This suggests that combining both perspectives may improve river health assessments. The findings offer a replicable framework that can help improve biomonitoring efforts in tropical freshwater systems.

## 1. Introduction

Traditionally, taxonomic approaches have been the primary method for assessing freshwater ecosystems, particularly by analyzing benthic macroinvertebrates across environmental gradients [1]. Many taxonomic metrics have emerged globally for evaluating these communities, including diversity metrics (e.g., richness, Margalef, Shannon) and pollution tolerance metrics, such as the widely used Biological Monitoring Working Party score (BMWP) [2]. Over the past 25 years, functional approaches have, however, gained prominence in benthic macroinvertebrate research [3]. Unlike taxonomic metrics, functional metrics are less specifically developed for aquatic communities; therefore, this research framework employs general ecological metrics originally designed for broader biodiversity studies [4]. These functional diversity metrics use biological traits (morphology, physiology, or behavior) linked to phylogeny to show how organisms interact with their environment [5]. For benthic macroinvertebrates, key traits include feeding habits, reproductive strategy, respiration type, and body morphology, among others [6,7]. According to several authors, incorporating evolutionary history through functional traits can reveal ecological patterns not captured by taxonomic analysis, offering deeper insights into responses to environmental gradients [8,9,10]. Given their complementary nature, numerous studies now apply both taxonomic and functional approaches to macroinvertebrate research [11]. Understanding how these two sets of metrics relate is key to improving predictions of freshwater ecosystem responses to disturbance and management actions [12,13]. Previous research, especially in fish communities, has typically explored this relationship through regression-based approaches [14,15,16], assessing the strength of correlation between pairs of diversity metrics. While regression-based approaches are useful for detecting direct statistical associations between diversity metrics, they may be less suitable for capturing the complex structure of ecological communities, particularly when multiple environmental processes operate simultaneously across heterogeneous gradients [17]. In contrast, this study uses an unsupervised clustering methodological approach to independently group sampling points according to each diversity metric. Clustering enables an integrative evaluation of community structure by grouping sites based on similarity patterns in each diversity metric, without requiring assumptions of linearity or external predictors. This enables a direct comparison of how taxonomic and functional diversity metrics reflect the structural variation in macroinvertebrate communities along a stream gradient. This study addresses the following research questions: (i) To what extent do taxonomic and functional diversity metrics generate similar sampling point groupings in a fluvial habitat context? (ii) Does the degree of concordance between taxonomic and functional diversity metrics vary along a gradient of fluvial habitat quality? To answer these questions, we apply a clustering-based framework to assess whether different diversity metrics (taxonomic and functional) lead to comparable sampling point groupings. We use the Measure of Concordance (MoC), a coefficient exclusively designed to quantify the agreement between clustering results, to evaluate the structural similarity between the groupings produced by each metric. Rather than assuming one set of diversity metrics to be superior, our overarching goal is to evaluate whether taxonomic and functional metrics offer redundant or complementary information when used to group macroinvertebrate communities. This framework contributes to a better understanding of the structural relationships between commonly used biodiversity metrics and could be useful for defining biomonitoring strategies under diverse ecological conditions. Our study area, the Paute River Basin, in southern Ecuador, is ecologically and economically strategic because of its high biodiversity and the key role that it plays in national hydroelectricity generation (approximately 40% of total hydroelectricity generation). The fluvial habitat index (FHI), based on physical habitat quality, is used as an environmental reference to assess how diversity patterns of macroinvertebrates relate to stream conditions.

## 2. Materials and Methods

### 2.1. Study Area

The Paute River Basin covers an area of 6442 km^2^ (Figure 1a). Land cover in the basin is heterogeneous, comprising intervened areas, native woody vegetation, páramo (high-altitude Andean ecosystem ranging in elevation from about 3000 to 5000 m), and urbanized zones (Figure 1b). The elevation ranges from 410 to 4687 m above sea level (m a.s.l.) (Figure 1c), with slope gradients varying between 25% and 50%. Lower average daily temperatures (approximately 6 °C) are recorded at elevations near 3500 m a.s.l. in the western Andes, while warmer temperatures (averaging 24 °C) are recorded in the Amazonian and subtropical valleys, where high diurnal temperature variation is also observed [18]. The basin exhibits strong altitudinal gradients in precipitation, with mean annual rainfall ranging from 660 mm in the central region to over 3400 mm near the outlet. High-elevation stations (above 3000 m a.s.l.) receive between 1000 and 1400 mm annually [19]. According to the 2010 national census, the largest cities in the basin are Cuenca (approximately 500,000 inhabitants) and Azogues (approximately 33,850 inhabitants).

### 2.2. Sampling Design

Data were collected from 12 sampling points distributed across the study basin (Figure 1a), collected by the former National Water Secretariat (SENAGUA) during the year 2008, the period of 2010–2013, and the year 2017. The dataset includes records for 51 families of benthic macroinvertebrates and the fluvial habitat index (FHI), based on the protocol of the US Environmental Protection Agency [20]. On average, each sampling point was visited five times per year, with increased sampling frequency at points identified as either impacted (mainly by human activities such as agriculture and channel alteration) or pristine (i.e., reference sites). Because of logistical constraints, monitoring in 2008 and 2017 was limited, resulting in fewer replicates for some sampling points during those years. In what follows, n_sp_ is the number of sampling points, n_rep_ the number of replicates, and n_v_ the number of variables. With n_sp_ = 12 sampling points, each revisited multiple times, the database contains n_rep_ = 216 sampling replicates. The data refer to n_fam_ = 51 families of benthic macroinvertebrates and n_FHI_ = one habitat quality index. The complete dataset encompasses n_v_ = 52 variables (n_v_ = n_fam_ + n_FHI_) and has a size of n_obs_ = 11,232 observations (n_obs_ = n_rep_ × n_v_). Although this study spans six years of monitoring, the relatively small number of sampling points (n_sp_ = 12) may be viewed as a limitation. Nevertheless, as shown in the Results section, no significant temporal or altitudinal variation was observed in the FHI values among replicates. This consistency likely reduces statistical noise from biogeographic factors, thus improving the robustness of the comparative analysis based on functional and taxonomic diversity.

#### 2.2.1. Benthic Macroinvertebrate Sampling

Sampling points were first established by defining a 20 m reach at each site. Within this reach, three evenly spaced transects were marked [21]. Following the definition of sampling point locations, the sampling campaigns were performed consistently for each sampling point and each was replicated across the study years. Macroinvertebrates were collected using a 25 × 25 cm^2^ nylon hand-net with a 0.5 mm mesh size, placed on the streambed, while the substrate was disturbed upstream by foot (kick sampling). Each three-minute sampling session covered all available microhabitats, differing in depth, substrate, and flow velocity. The three samples obtained along the transects were pooled to form a composite sample. A 20-min visual search of the substrate and aquatic vegetation was conducted to capture clinging taxa—such as Blephariceridae and Elmidae—that are often underrepresented in kick samples [22]. All macroinvertebrate samples were collected, preserved in 70% ethanol, and then transported to the laboratory where they were sorted using a stereomicroscope for taxonomic identification.

#### 2.2.2. The Fluvial Habitat Index (FHI)

Habitat quality was assessed for all the replicates of each sampling point using the fluvial habitat index (FHI) developed by the US Environmental Protection Agency (EPA) [20]. This index evaluates 13 habitat parameters: (1) epifaunal substrate/available cover, (2) embeddedness, (3) pool substrate characterization, (4) velocity/depth regimes, (5) pool variability, (6) sediment deposition, (7) channel flow status, (8) channel alteration, (9) riffle frequency (or channel bends), (10) channel sinuosity, (11) bank stability (left and right banks), (12) vegetative protection (left and right banks), and (13) riparian vegetative zone width (left and right banks). Each parameter is rated on a numeric scale, with higher scores reflecting better habitat quality. The individual scores are summed to yield a composite index, with a maximum value of 200. The FHI enables the distinction between different levels of stream quality by integrating ecological and biogeographical characteristics specific to a region. To ensure methodological consistency across all sites and sampling periods, FHI assessments were carried out by the same trained field team using standardized scoring sheets adapted from the EPA protocol. When multiple observers participated, joint calibration sessions were conducted at the beginning of each campaign to harmonize evaluation criteria and minimize inter-observer variability.

### 2.3. Evaluated Taxonomic Metrics

Four dimensionless metrics based on taxonomic (tx) aspects were calculated for all the replicates of each sampling point, i.e., the Shannon diversity (H), richness of families (N), the Margalef diversity (D_Mg_), and the Andean Biotic index (ABI). Detailed descriptions of H, N, and D_Mg_ can be found in the literature [23,24]. The ABI [25] is specifically tailored to Andean freshwater ecosystems above 2000 m a.s.l., encompassing all the sampling points considered in this study (Figure 1). It is derived from the BMWP index, is based on macroinvertebrate families, and assigns tolerance scores to each taxon, ranging from 10 for pollution-sensitive families to 1 for tolerant ones. The final ABI score is computed by summing the tolerance values of all families present in a sample.

### 2.4. Using Biological Traits of Macroinvertebrates into Functional Diversity Calculations

Eight biological traits were selected for this study (Table 1), encompassing forty-two functional macroinvertebrate categories. These traits and their corresponding categories are widely recognized as reliable indicators of ecological response to ecohydrological pressures [19,26]. A trait-by-family database (Db_f-trait_) was constructed for all the replicates of each sampling point, based on the affinity of each macroinvertebrate family to the respective functional categories. Affinity scores were assigned using a fuzzy coding procedure [27], following trait information compiled for the study basin [28]. These affinity values range from 0 (no affinity) to 3 (strong affinity). Values for the exoskeleton hardness trait category were determined based on expert judgment by the authors.

### 2.5. Functional Diversity Metrics

Using Db_f-trait_, four dimensionless functional diversity (FD) metrics were computed for each sampling point and its replicates, i.e., functional diversity based on dendrograms, including the taxa community pool (wFDc) [29]; functional dispersion (FDis) [30]; functional richness (FRic) [31]; and Rao’s quadratic diversity (Rao) [32]. Ward clustering with Euclidean distance was used to create dendrograms with FD metrics. Although several authors recommend species-level identification for diversity metrics, such as FD metrics [33], our approach aligns with a growing body of literature supporting the use of family-level identification in stream bio-assessments. Specifically, it was demonstrated that macroinvertebrate families are suitable surrogates for FD-based analyses in the evaluated basin [28], echoing findings from other studies [34,35,36,37,38]. This trend also applies to traditional taxonomic diversity metrics, such as Shannon and Margalef.

### 2.6. Rationale for Metric Selection

The rationale behind the selection of these metrics is twofold: balancing ecological relevance with methodological complementarity—that is, combining metrics that differ in how they quantify community structure (e.g., richness vs. evenness, or composition vs. sensitivity), providing a broader analytical framework. For taxonomic diversity, four widely used metrics were considered: H, which reflects both species richness and the balance in species abundances; D_Mg_, a richness-based index that adjusts the number of taxa according to sample size [23]; N, the number of benthic macroinvertebrate families (i.e., taxonomic richness); and ABI, which is relevant for high-altitude Andean streams. Unlike the other taxonomic metrics, ABI incorporates an explicit biotic evaluation based on pollution tolerance scores assigned to macroinvertebrate families. This adds a dimension of aquatic environmental quality assessment grounded in organismal sensitivity, making it highly relevant for biomonitoring in tropical Andean freshwater ecosystems. Together, these metrics capture a spectrum of taxonomic information, ranging from general diversity patterns to region-specific biotic responses to environmental disturbance. For functional diversity, the following metrics were selected: Rao, as it is the most widely applied metric in freshwater ecological communities [4]; wFDc, previously identified as particularly informative for the macroinvertebrates’ communities of the study basin [39]; FDis, which quantifies the dispersion of species in trait space and is mathematically related to Rao [30]; and FRic, which represents the volume of functional trait space occupied by a community [31]. Collectively, these metrics capture various aspects of functional diversity, ranging from trait redundancy to ecological breadth, enabling a more detailed and refined understanding of how communities respond to environmental gradients. This integrative set of metrics ensures both comparability with the existing literature and the exploration of distinct aspects of biodiversity relevant to environmental gradients and biomonitoring applications.

### 2.7. Data Analysis

#### 2.7.1. Clustering Sampling Points upon Their Fluvial Habitat Index Values

Cluster analysis aims to uncover intrinsic groupings within a dataset upon similarities among cluster members [40]. Non-hierarchical clustering techniques, such as K-means, are commonly used in freshwater ecology research [41], in which the number of clusters (nc) must be defined a priori. Cluster membership is determined by computing centroids for each cluster and assigning objects to the clusters with the nearest centroids. This assigning process is iteratively refined to minimize within-cluster dispersion [42]. In this study, the squared Euclidean distance ($D_{i,j}^{2}$) was used to assign objects to clusters, serving as the similarity measure to minimize within-cluster dispersion (Equation (1)):(1)$$D_{i,j}^{2}=\sum_{v}^{n_{v}}{\left(x_{i,v}-x_{j,v}\right)}^{2}$$
where x_i,v_ and x_j,v_ represent the values of variable *v* for objects *i* and *j*, respectively, with i ≠ j, and i, j < n_sp_. In this step of the analysis, clustering was performed using a single input variable—the fluvial habitat index (FHI) averaged per sampling point—thus setting n_v_ = 1. Under this condition, Equation (1) reduces to a basic squared distance between FHI scores.

For identifying nc, we employed an internal validity index, i.e., the Gap statistic [43,44,45], referred to as Gap_nc_, which compares the observed within-cluster dispersion to its expected value under a reference null distribution, providing an objective criterion for selecting the most appropriate nc value. We evaluated nc = 2, 3, 4, and 5, and selected the one yielding the highest Gap_nc_ value as the optimal number of clusters (nc_opt_). The K-means algorithm was then implemented to group the sampling points into clusters based on their FHI values. Following this, for a sampling point (out of the 12 study points), the respective FHI cluster identifier (${i.e.,FHI}_{C1},{FHI}_{C2},\ldots ,{FHI}_{{Cnc}_{opt}}$) was assigned to the original observations of that sampling point [46]. After completing this procedure for all 12 sampling points, the resulting dataset contained 11,232 observations, which were used in all subsequent analyses.

#### 2.7.2. Evaluation of Mathematical Concordance Between Taxonomic and Functional Metrics

To evaluate the level of agreement between taxonomic and functional diversity metrics, K-means clustering was independently applied to each of the eight study metrics (four taxonomic and four functional). For all clustering runs, nc was set to 3, following previous recommendations [28]. This process resulted in one partition (cluster assignment) per metric. To assess the similarity between taxonomic and functional metrics based on the clustering of replicate-level observations across sampling points, we conducted pairwise comparisons of the resulting clustering solutions. Unlike direct comparisons of index values or regression-based approaches, this clustering-based strategy allows for the detection of structural similarities in community composition patterns. Such an approach is suitable when community responses are shaped by multiple ecological processes acting across heterogeneous environmental gradients. To quantify the agreement between taxonomic and functional clusters, we employed the Measure of Concordance (MoC) [47]. The MoC is specifically designed to compare cluster partitions and provides a symmetric measure of structural overlap between two cluster outputs. It captures the extent to which groupings derived from different diversity metrics assign the same replicates to the same clusters, going beyond simple value-by-value correlation. Following the eight K-means procedures, pairwise comparisons between all functional and taxonomic cluster outputs were conducted using the MoC. Given that there are I clusters in K_fd-K_, and J clusters in K_tx-K_, and that each cluster in K_fd-K_ is referred to as fd_i_, and each cluster in K_tx-K_ as tx_j_, for i ∈ {1, 2,…, I} and j ∈ {1, 2,…, J}, then, any cluster fd_i_ can be subdivided into smaller subclusters. A subcluster (m_ij_) comprises those elements of fd_i_ that have also been allocated to a single cluster tx_j_; m_ij_ is, therefore, the intersection between fd_i_ and tx_j_. If cluster fd_i_ contains the subcluster m_ij_, then cluster tx_j_ also contains the same subcluster. The term $m_{ij}^{2}$/fd_i_ tx_j_ provides a symmetric measure of mutual concordance between the two clusters, fd_i_ and tx_j_. The maximum value $m_{ij}^{2}$/fd_i_ txd_j_ = 1 is attained only when fd_i_ = tx_j_. The MoC is calculated as follows and varies between 0 (no concordance) and 1 (perfect concordance):(2)$$MoC=\frac{\left(\sum_{i=1}^{I}\sum_{j=1}^{J}\frac{m_{ij}^{2}}{{fd}_{i}\;{txd}_{j}}-1\right)}{\left(\sqrt{IJ}-1\right)}$$

The choice of the Measure of Concordance (MoC) over other clustering comparison metrics was motivated by its conceptual alignment with our study goals. Unlike metrics such as the Adjusted Rand Index or Normalized Mutual Information, which often focus on element-level overlap or require distributional assumptions, MoC evaluates the structural similarity between partitions based on their global topological arrangement. This makes it especially suitable for ecological studies where clustering reflects emergent community structures rather than discrete, label-based classifications. Our intention was not to benchmark concordance metrics but to adopt one that preserves the structural logic of unsupervised clustering results while allowing for meaningful interpretation of functional–taxonomic alignment. In this context, MoC offered a parsimonious and interpretable solution [46].

While the MoC provides a global measure of clustering agreement, it does not show how individual sampling points and their replicates contribute to the overall similarity. To address this limitation, we complemented the MoC with a site-level analysis by calculating the percentage of cluster coincidence between functional and taxonomic metrics for each sampling point. Utilizing all available replicates per sampling point, this analysis was based on overlapping cluster assignments. This complementary approach allowed for a finer-scale interpretation of concordance beyond the global MoC value. This analysis was also performed across the fluvial habitat gradient to assess the consistency of community structure under varying environmental conditions.

Taxonomic metrics (H, N, and D_Mg_) were calculated using the PAST^®^ software version 5.1 [48], while functional metrics (wFDc, Rao, FRic, and FDis) were computed using the FDiversity^®^ software [49] (version 2008; download from https://fdiversity.dirienzo.com.ar/, accessed 10 March 2025). Computation of the Gap_nc_ statistic and execution of the K-means algorithm were implemented in MATLAB^®^ R2022a using customized subroutines. The ABI and MoC metrics were calculated using Microsoft Excel^®^. All input matrices and custom MATLAB^®^ scripts used in these analyses are included in the Appendix A to ensure full reproducibility.

## 3. Results

### 3.1. Outcomes of Clustering Sampling Points by Fluvial Habitat Index Values

Using the FHI average values, nc_opt_ was defined as being equal to 3 (the Gap_nc_ values were −0.68 for two clusters, 0.06 for three clusters, −0.09 for four clusters, and −0.06 for five clusters), which was used as a priori information for the respective K-means process that produced clusters FHI_C1_, FHI_C2_, and FHI_C3_ of the sampling points. Table 2 presents the distribution of sampling points across these clusters, along with a summary of hydro-geomorphological parameters. The identified groups follow a clear gradient in habitat quality: the mean FHI values were 166.9 for FHI_C1_, 133.8 for FHI_C2_, and 99.8 for FHI_C3_. Sampling points within each FHI cluster were also hydro-geomorphologically similar.

### 3.2. Clustering of Taxonomic Metrics as a Function of Fluvial Habitat

Figure 2 illustrates the distribution and clustering of the four taxonomic metrics, H, D_Mg_, N, and ABI. The left side of the figure shows the distribution of each metric (ordered from highest to lowest) across the sampling replicates. Vertical lines separate the three clusters generated for each metric through independent K-means analyses, based only on the internal distribution of taxonomic metrics values, without considering any habitat information. The right side of the figure presents pie charts that illustrate the distribution of fluvial habitat index (FHI) cluster labels, namely, FHI_C1_ (high-quality habitats), FHI_C2_ (intermediate), and FHI_C3_ (degraded), within each taxonomic metric cluster. Each replicate retained the FHI cluster label assigned to its corresponding sampling point, allowing us to evaluate alignment between taxonomic groupings (based on index values) and habitat quality groupings. A clear pattern emerged for most metrics: clusters associated with higher taxonomic index values (C1) were predominantly composed of replicates from FHI_C1_ sampling points, while clusters with lower values (C3) included an increasing proportion of replicates from FHI_C3_ points. This trend supported the sensitivity of taxonomic metrics to habitat degradation. In particular, H, D_Mg_, and ABI showed coherent structural transitions along the FHI gradient, as their K-means clusters shifted visibly from FHI_C1_ to FHI_C3_. In contrast, N exhibited greater internal heterogeneity across clusters, suggesting a weaker correspondence with habitat quality.

### 3.3. Clustering of Functional Diversity Metrics as a Function of Fluvial Habitat

Figure 3 illustrates the distribution and clustering of the four functional diversity metrics wFDc, Rao, FDis and FRic. The left side of the figure shows the distribution of each metric (ordered from highest to lowest) across the sampling replicates. Vertical lines separate the three clusters generated for each metric through independent K-means analyses, based exclusively on the internal distribution of functional diversity metrics values, considering no habitat information. The right side of the figure presents pie charts that illustrate the distribution of FHI cluster labels (FHI_C1_, FHI_C2_, and FHI_C3_), within each functional diversity metric cluster. Each replicate kept the FHI cluster label assigned to its corresponding sampling point, allowing us to evaluate alignment between functional diversity groupings (based on metric values) and habitat quality groupings. A clear pattern emerged for most metrics: clusters associated with higher functional diversity values (C1) were predominantly composed of replicates from FHI_C1_ sampling points, while clusters with lower values (C3) included an increasing proportion of replicates from FHI_C3_ points. This trend supported the sensitivity of functional diversity metrics to habitat degradation. In particular, wFDc, Rao, and FDis showed coherent structural transitions along the FHI gradient, as their K-means clusters shifted visibly from FHI_C1_ to FHI_C3_. In contrast, FRic exhibited greater internal heterogeneity across clusters, suggesting a weaker correspondence with habitat quality.

### 3.4. Mathematical Relationship Between Clusters from Taxonomic and Functional Metrics

The mathematical concordance between taxonomic and functional cluster outputs was evaluated using the MoC (Figure 4). The highest MoC values were observed for wFDc and FDis when paired with the taxonomic metrics H and D_Mg_, suggesting strong structural overlap between these taxonomic and functional perspectives. Rao showed moderate levels of agreement with most taxonomic metrics, while FRic consistently presented the lowest MoC values across all taxonomic metrics, not exceeding 0.5. These findings reinforced the idea that taxonomic metrics based on richness and relative abundance evenness (H and D_Mg_) align more closely with functional metrics that capture trait dispersion and dendrogram-based diversity. On the other hand, FRic, which quantifies the volume of occupied functional space, appeared less consistent with the patterns captured by the other metrics. This may reflect its distinct mathematical properties and its limited responsiveness to the trait-based clustering structure.

### 3.5. Relationship Between Taxonomic and Functional Diversity Clusters as a Function of Fluvial Habitat

In the final step, we evaluated the consistency between taxonomic and functional diversity clusters by calculating the percentage of cluster overlap for each sampling point using all available replicates. This analysis was conducted within the three FHI clusters (i.e., FHI_C1_, FHI_C2_, and FHI_C3_) to assess how environmental context influences the similarity in community structure across diversity perspectives. As shown in Figure 5, the level of agreement varied depending on the combination of metrics and the FHI cluster. For instance, combinations such as H with FDis or Rao showed consistently high overlap across all clusters, whereas other pairings, like FRic with ABI or N, reached higher agreement in transitional habitats (FHI_C2_). This variability suggests that the degree of structural concordance between taxonomic and functional diversity metrics is context-dependent and influenced by both the nature of the metrics and the ecological conditions of the sampling points. Among functional diversity metrics, FRic consistently showed the lowest agreement with taxonomic metrics across clusters. Similarly, N exhibited weak correspondence with most functional diversity clusters, especially when paired with trait-based metrics like Rao and FDis.

## 4. Discussion

Various metrics have been developed to assess anthropogenic impacts on lotic and lentic ecosystems, with benthic macroinvertebrates being one of the most widely used biological target groups. Traditionally, these assessments have focused more on taxonomic composition rather than on functional traits [1]. Recently, though, the trend has been changing toward using functional diversity (FD) metrics more often in freshwater ecology research [3,4,50,51], particularly in diverse, complex areas like the Andes [52], because including traits strengthens biomonitoring [53,54]. In this context, a growing concern (particularly in ecological modeling) is the potential redundancy among the many available diversity metrics [55]. Despite the widespread use of both functional and taxonomic diversity metrics [11,12,56], their mathematical concordance has rarely been examined, especially in benthic macroinvertebrates. This represents a significant gap in the literature, and exploring the overlap between taxonomic and functional diversity measures is necessary to improve the robustness and interpretability of biomonitoring tools. The present study addressed this gap by evaluating concordance between taxonomic and FD metrics applied to benthic macroinvertebrate communities across a fluvial habitat quality gradient.

### 4.1. Statistical Approach

FD is often assumed to have an asymptotic relationship with taxonomic richness. As more taxa are added to a community, the likelihood increases that new species will contribute to existing functional groups rather than introducing entirely new functions [57]. However, in systems under intermediate levels of disturbance, this asymptotic pattern may not be observed [58]. Previous research, especially on fish communities, has often relied on regression analyses to investigate the link between taxonomic and functional diversity metrics [14,15,16]; these approaches typically focused on direct associations between metric values. However, their ability to identify broad structural patterns may be limited, particularly in diverse environments influenced by various ecological factors [17]. The methodological approach implemented in this study addressed this limitation by applying, independently, unsupervised clustering (K-means) to each metric. This allowed for the emergence of community structure patterns based solely on the internal behavior of each metric. Once clusters are defined, they can be interpreted using any environmental framework (e.g., a fluvial habitat gradient). This flexible design enables a direct evaluation of how well different metrics responded to environmental variation, and to what extent taxonomic and functional perspectives produced congruent clusters under changing ecological conditions. The application of K-means clustering to each metric, separately, followed by pairwise concordance analysis using the Measure of Concordance (MoC), and replicate-level overlap, provided robust insights into how different metrics capture ecological patterns across habitat conditions. Although the MoC provided an overall estimate of clustering similarity between taxonomic and functional diversity metrics, this global index did not capture the contribution of individual sampling points to the observed concordance. Recent methodological developments highlight the importance of decomposing global clustering measures to better understand how local patterns contribute to overall agreement [59]. By calculating the percentage of cluster coincidence at the sampling point and replicate level, this study revealed finer-scale patterns of concordance that would otherwise remain hidden within a single global value. This two-tiered analytical approach not only strengthened the evaluation of metric congruence but also provided ecological insights into the consistency of community structuring along the fluvial habitat quality gradient. This approach went beyond traditional correlation-based comparisons and offered a replicable method for identifying structural convergence or divergence between taxonomic and trait-based metrics.

The findings reinforced the value of concordance-based metrics as a tool to evaluate the ecological coherence and complementarity of biodiversity metrics in biomonitoring. To our knowledge, this is the first study that combined unsupervised clustering of individual functional diversity metrics with formal concordance measures, such as MoC, to evaluate the structural agreement between taxonomic and functional diversity metrics. This framework offered a novel and replicable strategy for assessing index performance across ecological gradients.

### 4.2. Implications for Biomonitoring

Our results contributed to both (i) identifying redundant metrics and (ii) detecting metrics that capture distinct ecological dimensions. The replicate-level concordance analysis across the fluvial habitat gradient revealed that some combinations of taxonomic and functional diversity metrics consistently grouped sampling points similarly. For instance, the pairing H-FDis yielded high overlap values in all clusters (86.4% in FHI_C1_, 87.7% in FHI_C2_, and 81.9% in FHI_C3_), highlighting its stability and sensitivity across the entire gradient. Similarly, strong patterns were observed for H-Rao (83.8%, 82.3%, 82.2%) and D_Mg_-wFDc (85.0%, 85.9%, 85.5%). These results were consistent with the formal concordance analysis, as these same combinations also showed high values of MoC, confirming a strong structural alignment between the corresponding metrics across the FHI gradient (Figure 4). In contrast, FRic and N appeared less informative in relation to habitat quality (Figure 2c and Figure 3d), and were, therefore, not central to our interpretation. These findings were further supported by known relationships among certain metrics, e.g., Rao and FDis [30], which were also highly correlated with H in our dataset. An intriguing result was the consistently high structural concordance between D_Mg_ and wFDc across the fluvial habitat gradient (Figure 4 and Figure 5). Despite their distinct mathematical foundations—D_Mg_ being a richness-based taxonomic metric and wFDc a trait-based metric derived from hierarchical dendrograms weighted by taxa abundance—both metrics appear to capture similar community patterns. One plausible explanation is that, in these macroinvertebrate communities, taxonomic richness was closely linked to functional expansion: each additional taxon often contributed novel functional traits rather than being functionally redundant [60]. This relationship was likely in structurally complex habitats (e.g., FHI_C1_), where niche differentiation is stronger [61,62]. In such an environment, the increase in taxonomic diversity may directly reflect an increase in functional diversity, resulting in a parallel response from both metrics. The hierarchical nature of both metrics—Margalef responding to richness scaled by sampling effort, and wFDc reflecting functional spread within a structured dendrogram—may further contribute to their structural alignment [63,64]. This finding highlighted the importance of not assuming functional–taxonomic divergence solely based on methodological differences, as, under certain ecological conditions, even mathematically distinct metrics can yield convergent insights. Importantly, our results also challenged the assumption that structural alignment among metrics is always desirable or indicative of ecological relevance. For example, ABI displayed moderate sensitivity to the FHI gradient (Figure 2d), confirming its ecological informativeness, yet showed lower concordance with functional metrics. This divergence suggests that ABI captures unique aspects of biological integrity not reflected in trait-based metrics. This nuanced perspective reinforced the growing concern (particularly in ecological modeling) regarding potential redundancy among diversity metrics. Rather than selecting metrics solely based on convention or availability, biomonitoring frameworks should strategically combine both structurally concordant metrics and ecologically informative divergent ones: the former to detect robust, generalizable trends, and the latter to reveal complementary signals of ecosystem structure and change. This approach offers a replicable pathway for disentangling structural similarity from ecological signal strength in community diversity studies. These findings are consistent with recent studies on benthic macroinvertebrates that have challenged the assumption of functional metrics being inherently more informative. For example, Liu et al. [11] reported that taxonomic diversity metrics showed higher sensitivity to distinguish different types of land-use compared to functional diversity measures. Similarly, a study on tropical river systems found that the strengths of trait–environment relationships were lower than taxon–environment relationships [65]. In shallow lake ecosystems, Zhang et al. [66] observed a stronger trend toward taxonomic homogenization than functional homogenization under nutrient enrichment, further showing that taxonomic perspectives can be more responsive to certain environmental gradients. Even outside aquatic systems, divergence between taxonomic and functional responses has been observed—such as in Tibetan alpine grasslands, where fertilization decreased species diversity but increased functional diversity [67]. Together, these studies support the idea that low concordance does not imply irrelevance, but rather ecological complementarity.

## 5. Conclusions

This study showed that taxonomic and functional diversity (FD) metrics, when analyzed through a concordance-based framework, provided consistent yet complementary views of benthic macroinvertebrate community structure along a fluvial habitat quality gradient. High levels of agreement were found between taxonomic metrics that reflect relative abundance evenness and richness (i.e., H and D_Mg_) and functional metrics that capture either trait dispersion (i.e., FDis, Rao) or the structure of community-weighted dendrograms (i.e., wFDc). In contrast, FRic and N exhibited lower levels of concordance and were also less informative regarding the fluvial habitat gradient. Using clustering coincidence and the Measure of Concordance (MoC) provided a novel and replicable approach to assess the structural alignment between biodiversity perspectives. Importantly, this study highlighted that low concordance between metrics does not imply ecological irrelevance. For instance, the ABI strongly responded to the fluvial habitat gradient but showed weak alignment with functional metrics, showing that it captures unique aspects of biological integrity beyond trait-based or taxonomic structure. Integrating functional trait-based information with traditional taxonomic assessments could enhance the robustness of biomonitoring programs by enabling the detection of both broad and subtle ecological signals. Strategic selection of metrics should therefore consider not only their internal structural concordance but also their independent ecological informativeness. While the analytical framework applied in this study offers a robust approach for evaluating the concordance between taxonomic and functional diversity metrics, its applicability may be influenced by the specific ecological and geographic context of the study site. The combination of clustering and concordance analysis provides a transferable method that can be adapted to other freshwater systems. Although the patterns observed may reflect localized ecological dynamics, they offer a replicable structure for exploring how different diversity metrics align in response to environmental gradients.

## Figures and Tables

**Figure 1 biology-14-00692-f001:**
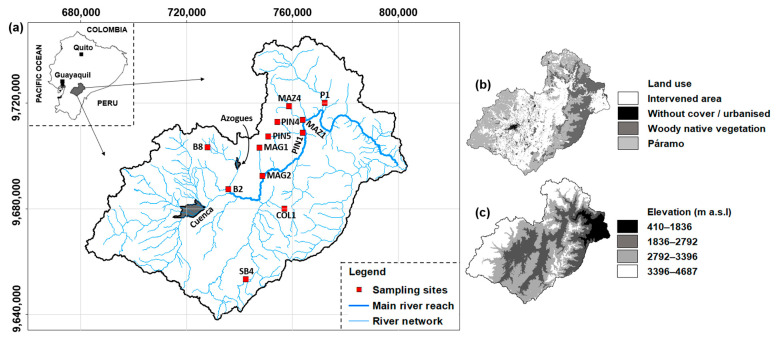
(**a**) Location of the Paute River Basin in continental Ecuador, including the cities of Cuenca and Azogues, and spatial distribution of the twelve sampling points. (**b**) Land use categories and (**c**) elevation categories throughout the basin. Coordinates’ system: WGS84 UTM Zone 17S (meters).

**Figure 2 biology-14-00692-f002:**
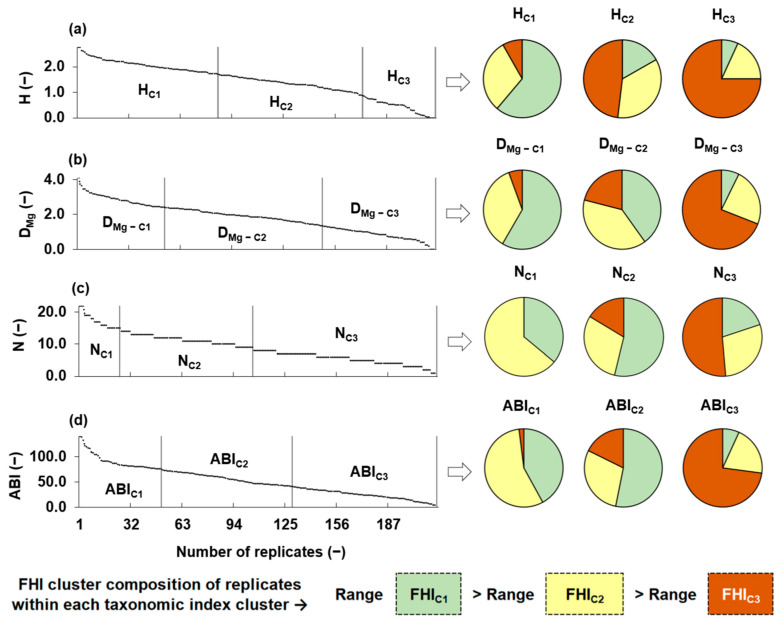
Distribution of taxonomic metrics calculated for the replicates observed at the 12 sampling points, and proportion (i.e., pie charts) of fluvial habitat index (FHI) cluster labels within each taxonomic metric cluster. Taxonomic metrics are as follows: (**a**) Shannon diversity (H), (**b**) Margalef diversity (DMg), (**c**) family richness (N), and (**d**) Andean Biotic Index (ABI). Vertical lines separate the clusters obtained independently for each taxonomic metric based on K-means analysis. Pie charts show the composition of each taxonomic cluster in terms of the FHI classification: green = FHI_C1_, yellow = FHI_C2_, and orange = FHI_C3_. The lower section summarizes the FHI cluster composition of replicates within each taxonomic metric cluster and highlights that the range of variation in FHI values decreases along the following gradient: range(FHI_C1_) > range(FHI_C2_) > range(FHI_C3_). Here, range(FHI) refers to the spread of FHI values within each respective habitat quality cluster.

**Figure 3 biology-14-00692-f003:**
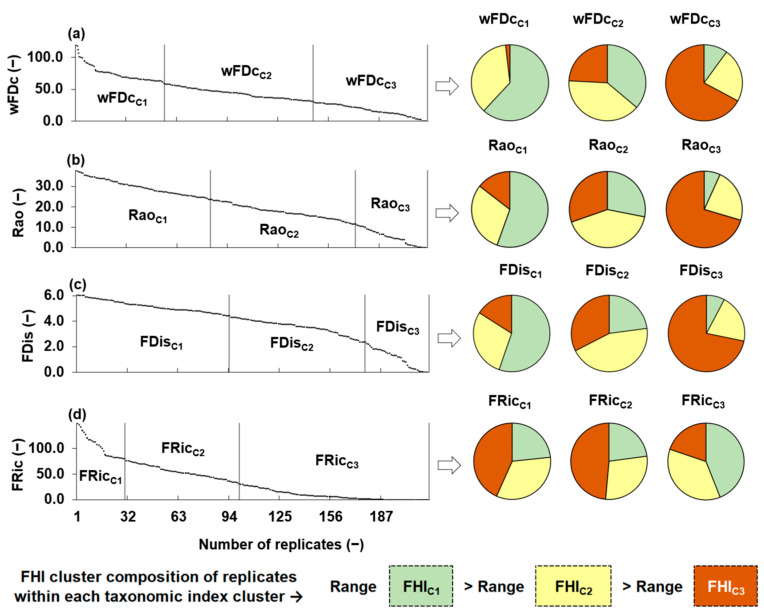
Distribution of functional diversity metrics calculated for benthic macroinvertebrate communities at 12 sampling points and their replicates. These include the following: (**a**) wFDc = functional diversity based on dendrogram topology and taxa community pool, (**b**) Rao = Rao’s quadratic entropy, (**c**) FDis = functional dispersion, and (**d**) FRic = functional richness. Vertical lines separate the K-means clusters obtained independently for each metric. Pie charts on the right display the composition of each functional metric cluster in terms of the fluvial habitat index (FHI) classification: green = FHI_C1_, yellow = FHI_C2_, and orange = FHI_C3_. The lower section summarizes the FHI cluster composition of replicates within each functional metric cluster and highlights that the range of variation in FHI values decreases across the following gradient: range(FHI_C1_) > range(FHI_C2_) > range(FHI_C2_). Here, range(FHI) refers to the spread of values within each respective habitat quality cluster.

**Figure 4 biology-14-00692-f004:**
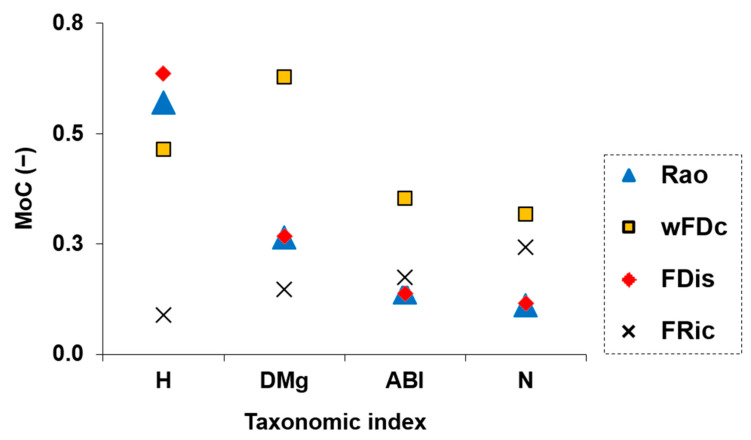
Pairwise concordance between taxonomic and functional diversity metrics, evaluated using the Measure of Concordance (MoC). Each point represents the MoC value obtained by comparing the clustering structure of one taxonomic metric with one functional metric. Taxonomic metrics include H = Shannon diversity, D_Mg_ = Margalef diversity, N = family richness, and ABI = Andean Biotic Index. Functional metrics include wFDc = dendrogram-based diversity (including the taxa community pool), Rao = Rao’s quadratic entropy, FDis = functional dispersion, and FRic = functional richness.

**Figure 5 biology-14-00692-f005:**
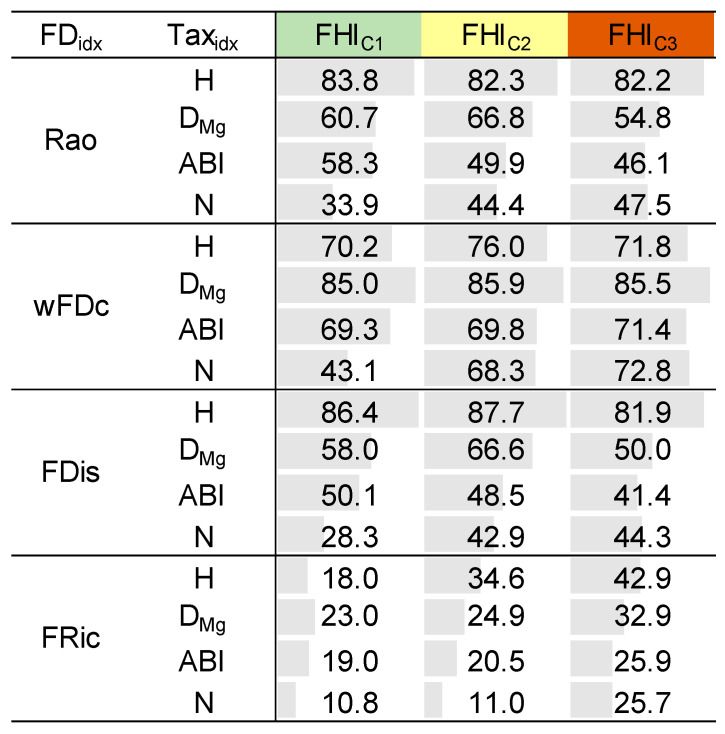
Percentage of cluster assignment coincidences between functional diversity (FD_idx_) and taxonomic (Tax_idx_) metrics, calculated for the replicates of each sampling point and summarized by the fluvial habitat index (FHI) cluster. H = Shannon diversity, D_Mg_ = Margalef diversity, N = family richness, ABI = Andean Biotic Index, Rao = Rao’s quadratic entropy, wFDc = functional dendrogram-based diversity, FDis = functional dispersion, and FRic = functional richness.

**Table 1 biology-14-00692-t001:** Functional traits and categories used in this study.

Trait	Category
Feeding habits	Collector–Filterer (C-Ft)
	Collector–Gatherer (CG)
	Piercers (Pc)
	Predators (Pr)
	Scrapers (Sc)
	Shredders (Sh)
	Parasite (PA)
Respiration	Tegument (Teg)
	Gill
	Plastron (Pla)
	Spiracle (Spi)
Body form	Streamlined (Str)
	Flattened (Flat)
	Cylindrical (Cy)
	Spherical (Sph)
Maximum body size (mm)	<2.5
	2.5–5.0
	5–10
	10–20
	20–40
	40–80
	>80
Body flexibility	None (<10°)
	Low (10°–45°)
	High (>45°)
Locomotion	Flier (Fli)
	Surface swimmer (SS)
	Full water swimmer (FWS)
	Crawler (Cra)
	Burrower (Bur)
	Temporarily attached (TA)
Reproduction	Asexual (As)
	Clutches and cemented (CC)
	Clutches and free (CF)
	Clutches in vegetation (CV)
	Clutches and Terrestrial (CT)
	Isolated eggs and clutches (IEC)
	Isolated eggs and free (IEF)
	Ovoviviparity (Ovi)
Hardness exoskeleton	None
	Moderate
	High

**Table 2 biology-14-00692-t002:** Clusters of sampling points according to their fluvial habitat index (FHI) values. Average hydro-geomorphological parameters are provided for each cluster, along with their respective ± standard deviations. ANuR = average number replicates; a.s.l. = above sea level; and Strahler = river order.

Sampling Point	FHI		ANuR	Elevation	Slope	Strahler
	(−)	Cluster	(−)	(m a.s.l.)	(%)	(−)
P1	173 ± 2.2	FHI_C1_	19	2855.3 ± 500.8	9.3 ± 11.4	3 ± 1.2
COL1	171 ± 2.5					
SB4	169 ± 1.1					
PIN5	154 ± 1.7					
MAZ4	137 ± 1.0	FHI_C2_	18	2384.1 ± 359.1	5.2 ± 3.7	4 ± 0.8
PIN4	135 ± 1.9					
PIN1	134 ± 3.2					
MAZ1	130 ± 2.4					
B8	107 ± 2.3	FHI_C3_	18	2611.5 ± 419.0	11.5 ± 13.4	4 ± 1.7
MAG2	105 ± 1.2					
B2	96 ± 1.5					
MAG1	91 ± 2.2					

## Data Availability

The raw macroinvertebrate data and functional trait information used in this study are provided as Appendix A. A matrix formatted for use in the FDiversity^®^ software is also included to enable replication of functional diversity calculations. Additionally, the MATLAB^®^ scripts used for Gapnc statistic computation and K-means clustering are provided.

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
