# Peer review of "Evaluating Mathematical Concordance Between Taxonomic and Functional Diversity Metrics in Benthic Macroinvertebrate Communities"

_biology, 2025, doi:10.3390/biology14060692_

Round 1

Reviewer 1 Report

Comments and Suggestions for Authors

Review for the paper “Evaluating mathematical concordance between taxonomic and functional diversity metrics in benthic macroinvertebrate communities” by Gonzalo Sotomayor and co-authors submitted to “Biology”.

The authors of this research paper conducted an analysis of the relationship between taxonomic and functional diversity metrics within benthic macroinvertebrate communities in the Paute River Basin, Ecuador based of macroinvertebrate samples from multiple locations along the river. The authors found that most of the taxonomic indices, with the exception of family richness and functional richness, responded positively to the fluvial habitat index. The study emphasizes the potential benefits of utilizing a multi-faceted approach that combines taxonomic and trait-based metrics, particularly in ecosystems facing ecological transitions or habitat degradation. By incorporating diverse indices, researchers and practitioners can improve the robustness of assessments aimed at understanding and managing freshwater biodiversity. Consequently, the findings of this study may have important implications for the integration of taxonomic and functional indices in freshwater biomonitoring frameworks.

The paper is well organized and coherently written. Only minor improvements are required.

Recommendations.

A simple summary is missing from the paper. This mandatory section should be included in the text.

Introduction.

L 52-54. The authors should explain the limitations of traditional taxonomic approaches that functional diversity metrics aim to address, especially in freshwater ecosystems.

L 84. The authors should explain why the Measure of Concordance (MoC) is an appropriate method for evaluating structural similarity between clustering results. Why was this method chosen over other similarity measures?

Materials and Methods.

L 100. The authors should define the "Páramo" area.

L 122–123: What criteria were used to classify sampling sites as "impacted" or "pristine"?

L 144-146: Given the variability in stream substrate, flow velocity, and aquatic vegetation, the authors should report how visual searches were standardized across sites to ensure methodological consistency.

Table 1. Consider replacing "Llocomotion" with "Locomotion."

Results.

L 288. It would be useful to compare the community structure within each cluster using PERMANOVA and to analyze how this structure relates to the results obtained from the clusters

L 294-295. It would also be useful to statistically compare these mean values using an appropriate test.

L 320-321, 350-351. To support the trends indicated in Figures 2 and 3, it would be useful to compute the average values for each index and compare them between clusters using an ANOVA or a Kruskal–Wallis test.

Discussion.

The authors should discuss specific sampling points where taxonomic-functional coherence was particularly high or low, as well as the factors that might explain these results.

They should also discuss the specific ecological or environmental conditions that caused certain metric combinations (e.g., H-FDis and DMg-wFDc) to yield high concordance values across the fluvial habitat gradient.

Although the supplementary material is provided, it is not cited. The authors should include the appropriate citations.

Author Response

C1: The authors of this research paper conducted an analysis of the relationship between taxonomic and functional diversity metrics within benthic macroinvertebrate communities in the Paute River Basin, Ecuador based of macroinvertebrate samples from multiple locations along the river. The authors found that most of the taxonomic indices, with the exception of family richness and functional richness, responded positively to the fluvial habitat index. The study emphasizes the potential benefits of utilizing a multi-faceted approach that combines taxonomic and trait-based metrics, particularly in ecosystems facing ecological transitions or habitat degradation. By incorporating diverse indices, researchers and practitioners can improve the robustness of assessments aimed at understanding and managing freshwater biodiversity. Consequently, the findings of this study may have important implications for the integration of taxonomic and functional indices in freshwater biomonitoring frameworks.

A1: We sincerely thank Reviewer 1 (R1) for his/her positive summary of the manuscript and for recognizing the value of integrating taxonomic and functional diversity metrics in freshwater biomonitoring. We are glad that the potential implications of this approach were clearly perceived.

C2: The paper is well organized and coherently written. Only minor improvements are required.

A2: We thank R1 for his/her kind assessment on the quality of our manuscript. We have thoroughly revised the manuscript to enhance its clarity and consistency.

C3: A simple summary is missing from the paper. This mandatory section should be included in the text.

A3: We thank R1 for pointing out the omission of the Simple Summary. Following the journal’s requirements, we have now included a Simple Summary at the beginning of the manuscript (lines 19-29).

C4: L 52-54. The authors should explain the limitations of traditional taxonomic approaches that functional diversity metrics aim to address, especially in freshwater ecosystems.

A4: While we agree with the importance of clarifying the limitations of traditional taxonomic approaches, we believe this explanation is best placed within the Introduction, where it is already addressed in detail (Lines 70-73). Due to space constraints and the need to maintain clarity in the abstract, we decided not elaborating further on this issue in this section of the manuscript. Nevertheless, if the editor deems it necessary, we would be happy to adjust the abstract to include this explanation.

C5: L 84. The authors should explain why the Measure of Concordance (MoC) is an appropriate method for evaluating structural similarity between clustering results. Why was this method chosen over other similarity measures?

A5: We thank the reviewer for the comment. While the rationale for selecting the Measure of Concordance (MoC) was initially introduced in the Methods (Section 2.7.2), we now clarify in the revised manuscript why MoC was chosen over other similarity metrics, such as the Adjusted Rand Index (ARI) or Normalized Mutual Information (NMI). As explained in the newly added paragraph in the Discussion (Section 4.1), MoC evaluates structural similarity between clustering partitions based on their topological configuration rather than element-wise overlaps or probabilistic assumptions. This structural focus aligns more closely with our study goals, which center on identifying emergent community patterns in functional and taxonomic diversity, rather than assessing label-level agreement. Our intention was not to benchmark clustering similarity indices, but to select one that preserves the logic of unsupervised clustering while offering meaningful ecological interpretation. We appreciate the reviewer’s suggestion, which helped us to strengthen the clarity and transparency of our methodological justification.

C6: L 100. The authors should define the "Páramo" area.

A6: We agree that the term “Páramo” may not be familiar to all readers. We have therefore added a short explanatory note in parentheses to clarify that it refers to a high-altitude Andean grassland ecosystem (L111-112).

 C7: L 122–123: What criteria were used to classify sampling sites as "impacted" or "pristine"?

A7: The classification of sampling sites as “impacted” or “pristine” was based on field observations and evident anthropogenic pressures, such as the presence of agriculture, infrastructure, and channel modifications. Pristine sites corresponded to reference points without observable signs of disturbance. Given the descriptive nature of this distinction, we did not include additional classification criteria in the manuscript, but we would be happy to provide more details if needed.

C8: L 144-146: Given the variability in stream substrate, flow velocity, and aquatic vegetation, the authors should report how visual searches were standardized across sites to ensure methodological consistency.

A8: We thank the reviewer for this valuable observation. We agree that the original response did not clearly explain how visual assessments were standardized across sites when applying the Fluvial Habitat Index (FHI). To address this, we have now incorporated a clarification at the end of Section 2.2.2 in the revised manuscript. Specifically, we state that all FHI evaluations were conducted by the same trained field team using standardized scoring sheets adapted from the EPA protocol. In cases where more than one observer was involved, joint calibration sessions were carried out at the beginning of each field campaign to harmonize scoring criteria and reduce inter-observer variability. This ensures consistency and comparability of visual assessments across sites with varying environmental conditions.

C9: Table 1. Consider replacing "Llocomotion" with "Locomotion."

A9: We thank R1 for noticing this typographical error. We have corrected it in Table 1.

C10: L 288. It would be useful to compare the community structure within each cluster using PERMANOVA and to analyze how this structure relates to the results obtained from the clusters

A10: We thank R1 for this thoughtful suggestion. However, we respectfully consider that applying a PERMANOVA to compare community structure within each cluster falls outside the primary scope of this study. Our objective is not validating the ecological distinctiveness of the clusters per se, but rather evaluating the structural concordance between taxonomic and functional diversity metrics based on their clustering outcomes. While a PERMANOVA-based analysis could be useful for future ecological validation, our current focus is on methodological alignment between indices rather than on the direct composition of the communities. That said, we truly appreciate this insight and will consider incorporating such community-level comparisons as part of a follow-up study focused on ecological interpretation and validation of metric-based groupings.

C11: L 294-295. It would also be useful to statistically compare these mean values using an appropriate test.

A11: We appreciate the reviewer’s suggestion and the opportunity to clarify our rationale. However, we respectfully maintain that performing a separate statistical test (e.g., ANOVA or Kruskal-Wallis) to compare FHI values across the clusters would be methodologically redundant. This is because the clusters were generated using K-means applied directly to FHI values, meaning that group differentiation is inherently based on this variable. Conducting additional statistical tests on a variable already used to define the groups would violate the assumption of independence and result in a tautological analysis. Moreover, the clustering procedure was not arbitrary—it was internally validated using the Gap Statistic, a robust method for determining the optimal number of clusters by comparing within-cluster dispersion against a reference null distribution. This internal validation reinforces the reliability and structural relevance of the resulting clusters. In the Results section, we also provide detailed descriptive statistics (means, medians, ranges) to support the ecological interpretation of differences among clusters. We believe this approach is methodologically sound, statistically justified, and aligned with the core principles of unsupervised clustering analysis.

C12: L 320-321, 350-351. To support the trends indicated in Figures 2 and 3, it would be useful to compute the average values for each index and compare them between clusters using an ANOVA or a Kruskal–Wallis test.

A12: We thank R1 for this follow-up suggestion. However, as with the previous comment (C11), we respectfully consider that conducting ANOVA or Kruskal–Wallis tests on the diversity index values across clusters would be methodologically redundant. The clusters shown in Figures 2 and 3 were formed using the K-means algorithm applied directly to those same index values. Therefore, differences between cluster means are inherently expected and are already visually represented through the ranked distribution plots and the corresponding FHI composition pie charts. Nonetheless, to provide a complete and respectful response to this reviewer’s suggestion, we conducted Kruskal–Wallis tests to assess statistical differences in diversity index values between clusters for each taxonomic and functional metric. As anticipated, significant differences were found between all clusters across all indices. Specifically, for the indices H, ABI, Margalef, richness, wFDc, Rao, FDis, and FRic, the Kruskal–Wallis tests yielded p-values of 0, confirming strong differentiation. This outcome reinforces the logic of the clustering approach and supports our initial hypothesis that additional post-clustering statistical tests would likely be redundant. It is important to note that these tests were performed solely to address this reviewer’s comment and are not included in the manuscript, as we maintain that the K-means clustering method already provides a valid and interpretable structure based on the diversity metrics themselves. As such, further statistical comparison of cluster means does not offer independent or novel insights for inclusion in the manuscript.

C13: The authors should discuss specific sampling points where taxonomic-functional coherence was particularly high or low, as well as the factors that might explain these results.

A13: We respectfully consider that discussing individual sampling points would introduce a level of local detail that goes beyond the scope and purpose of this study. Our objective is not to interpret site-specific drivers of diversity, but rather to evaluate the overall structural concordance between functional and taxonomic diversity indices in a way that is generalizable and methodologically informative. We do include a synthesis of concordance patterns by FHI cluster (Table 3), which provides a more interpretable summary of the relationship between diversity metrics and habitat quality without focusing on local conditions. We believe that this approach is more aligned with the expectations of a broad, international readership and with the intended contribution of our study to comparative biodiversity assessment frameworks.

C14: They should also discuss the specific ecological or environmental conditions that caused certain metric combinations (e.g., H-FDis and DMg-wFDc) to yield high concordance values across the fluvial habitat gradient.

A14: We would like to point to the fact that the discussion already addresses this aspect in detail. Specifically, we explain that the high concordance observed between DMg and wFDc likely results from structurally complex habitats (e.g., FHIC1) where niche differentiation is strong, leading to functional expansion with each added taxon. For the combination H–FDis, we discuss both the sensitivity of H across the gradient and its known association with functional dispersion. These interpretations are found in Section 4.5 and are supported by ecological reasoning and literature references. We believe this addresses the reviewer’s suggestion.

C15: Although the supplementary material is provided, it is not cited. The authors should include the appropriate citations.

A15: The authors would like to thank R1 for this and other constructive corrections. We have now added explicit references to the Supplementary Material in the Methods section, indicating that all matrices and scripts used in the calculations are provided to support reproducibility.

Reviewer 2 Report

Comments and Suggestions for Authors

Well written and informative analysis. This manuscript makes an important contribution to the field of bio-monitoring and the use of different indices and the relationship between taxonomic and functional indices and habitat (fluvial habitat).  The authors make a strong case for integrating functional trait-based information and taxonomic assessments to enhance bio-monitoring programs.

The only minor error/typo was in table 1: Llocomotion should be Locomotion.

Author Response

C1: Well written and informative analysis. This manuscript makes an important contribution to the field of bio-monitoring and the use of different indices and the relationship between taxonomic and functional indices and habitat (fluvial habitat).  The authors make a strong case for integrating functional trait-based information and taxonomic assessments to enhance bio-monitoring programs.

A1: The authors would like to thank Reviewer 2 (R2) for his/her positive evaluation of our manuscript and for recognizing the relevance of our contribution to biomonitoring through the integration of taxonomic and functional diversity metrics. We really appreciate the encouraging feedback.

C2: The only minor error/typo was in table 1: Llocomotion should be Locomotion.

A2: We thank R2 for catching this typographical error. This amendment was carried out in Table 1.

Reviewer 3 Report

Comments and Suggestions for Authors

Although the Authors utilise real-life data, the reviewed paper is a purely theoretical one. The Authors use K-means clustering to assign watercourses of the Paute River basin into three categories. The analysis was repeated thrice, and each time was based on different set of abiotic and biotic characteristics. Finally, the concordance of the classifications was tested. The strongest point of the paper is, in my opinion, showing that both indices of taxonomic diversity, and indices of functional diversity similarly classify levels of anthropogenic impact, and it is worthwhile to adopt both approaches in biomonitoring programmes.

The paper is definitely worth publishing, I would suggest, however, attending to some points to make it more “palatable”:

  1. Many of the abbreviations invented by the Authors pop up in the text far from the place where they were explained thus forcing a reader to go back and search. Perhaps it would be possible to repeat names of values/parameters more frequently?

  2. Sampling sites are not sufficiently characterised. A table with the raw values of parameters used for calculating the FHI would be handy (maybe included in the supplementary material).

  3. Using ASPT derived from the ABI, rather than simple ABI sum, may make a more precise index.

  4. Indices discussed in the paper do not assess the quality of water – they assess the quality of aquatic environment. Water quality is being assessed at potable water treatment plants.

  5. In the Table 2, Strahler river order appears. Using this parameter should be mentioned in the Methods section.

  6. Figure 1 is too small.

    1. Using geographic co-ordinate system would make the map clearer.

  7. Table 1:

    1. Locomotion instead of Llocomotion.

    2. “Water column swimmer” may sound better than “Full water swimmer”

    3. I have some doubts about “hardness [of] exoskeleton”. This parameter changes during the lifetime of an individual. Which point in time have the Authors taken into consideration?

Author Response

C1: Although the Authors utilise real-life data, the reviewed paper is a purely theoretical one. The Authors use K-means clustering to assign watercourses of the Paute River basin into three categories. The analysis was repeated thrice, and each time was based on different set of abiotic and biotic characteristics. Finally, the concordance of the classifications was tested. The strongest point of the paper is, in my opinion, showing that both indices of taxonomic diversity, and indices of functional diversity similarly classify levels of anthropogenic impact, and it is worthwhile to adopt both approaches in biomonitoring programmes.

A1: We thank Reviewer 3 (R3) for their careful reading and positive assessment of our manuscript. We would like to clarify that, while the study is indeed methodological in design, it is grounded in real-life empirical data collected across multiple years and sites in the Paute River Basin. The analytical framework we propose—including clustering and concordance assessment—was applied to actual community-level biodiversity data and habitat quality indices. We are pleased that the reviewer recognizes the value of integrating both taxonomic and functional indices, and we hope this contribution supports future biomonitoring efforts.

C2: The paper is definitely worth publishing, I would suggest, however, attending to some points to make it more “palatable”:

A2: We thank R3 for his/her overall positive evaluation and for the constructive spirit of his/her suggestions. We have carefully addressed the specific points raised by R3 to improve the clarity of the manuscript.

C3: Many of the abbreviations invented by the Authors pop up in the text far from the place where they were explained thus forcing a reader to go back and search. Perhaps it would be possible to repeat names of values/parameters more frequently?

A3: The authors agree with the suggestion of R3. Thus, whenever we believed it necessary, we have repeated once or twice the names of all indices in the newer version of the manuscript.

C4: Sampling sites are not sufficiently characterised. A table with the raw values of parameters used for calculating the FHI would be handy (maybe included in the supplementary material).

A4: The fluvial habitat index (FHI) used in this study was applied as an aggregated habitat quality score, not as a variable to be deconstructed or recalculated. The individual raw values from field forms used to calculate the FHI were not included, as they are not essential to reproduce or interpret the analyses presented. Furthermore, it is not standard practice to include this level of detail in the supplementary material for studies that rely on previously established indices. We have ensured, however, that the interpretation and use of the FHI in our study are clearly described.

C5: Using ASPT derived from the ABI, rather than simple ABI sum, may make a more precise index.

A5: While we acknowledge that ASPT-type transformations can offer useful alternatives in some contexts, we intentionally chose to use the Andean Biotic Index (ABI) in its standard form due to its widespread application and regional relevance for Andean freshwater ecosystems. Using an ASPT derived from the ABI would result in a very similar pattern, offering limited additional insight while potentially diluting the interpretation of an index that is already widely used and understood in regional biomonitoring frameworks.

C6: Indices discussed in the paper do not assess the quality of water – they assess the quality of aquatic environment. Water quality is being assessed at potable water treatment plants.

A6: The authors agree with R3 in that the indices used in this study—particularly the ABI—do not assess water quality in the physicochemical sense, but rather reflect the ecological condition of the aquatic environment through organismal tolerance to pollution. To avoid confusion, we have revised the terminology in the relevant sentence, replacing “water quality” with “aquatic environmental quality.” This better reflects the biotic and ecological scope of the index and improves alignment with the manuscript’s focus on habitat-based biomonitoring.

C7: In the Table 2, Strahler river order appears. Using this parameter should be mentioned in the Methods section.

A7: The Strahler river order appears in Table 2 solely as a descriptive variable to help contextualize the sampling points. It was not used in any analytical step or metric computation. As such, we have not included it in the Methods section. However, we have clarified its purpose in the table caption to avoid confusion.

C8: Figure 1 is too small. Using geographic co-ordinate system would make the map clearer.

A8: We respectfully note that Figure 1 was designed to clearly convey the spatial context, land use categories, and elevation structure of the Paute River Basin. The figure includes a full legend, coordinate reference system (WGS84 UTM Zone 17S), and appropriate visual elements to support interpretation. Given these features, we believe the current size and format are adequate for clarity and readability.

C9: Locomotion instead of Llocomotion.

A9: The authors would like to thank R3 for spotting this typographical error. This was amended in Table 1.

C10: “Water column swimmer” may sound better than “Full water swimmer”

A10: The term “Full Water Swimmer (FWS)” was intentionally used to maintain consistency with previous studies and trait databases that apply this classification. While “Water Column Swimmer” may also be a valid alternative, the authors would prefer to retain “FWS” for clarity and continuity in the context of our functional trait framework.

C11: I have some doubts about “hardness [of] exoskeleton”. This parameter changes during the lifetime of an individual. Which point in time have the Authors taken into consideration?

A11: We clarify that functional traits, including “exoskeleton hardness,” were assigned based on the ecological characteristics of macroinvertebrates at their aquatic life stage — typically the larval stage for insect taxa. In freshwater biomonitoring, the larval forms are the primary focus, as they are the life stage sampled in benthic communities and are most representative of ecological strategies in lotic systems. For example, larval stages of Elmidae (riffle beetles) already exhibit a strongly sclerotized exoskeleton, while Ephemeroptera nymphs have softer exoskeletons. These trait assignments reflect functional adaptations to aquatic environments and are based on consistent, taxonomically informed trait databases. Therefore, although trait expression may vary across life stages, our analysis focuses on the aquatic forms actually present in the benthic samples, ensuring ecological relevance and consistency.